# Optogenetic Brain–Computer Interfaces

**DOI:** 10.3390/bioengineering11080821

**Published:** 2024-08-12

**Authors:** Feifang Tang, Feiyang Yan, Yushan Zhong, Jinqian Li, Hui Gong, Xiangning Li

**Affiliations:** 1Britton Chance Center for Biomedical Photonics, Wuhan National Laboratory for Optoelectronics, MoE Key Laboratory for Biomedical Photonics, Huazhong University of Science and Technology, Wuhan 430074, China; tangfeifang@hust.edu.cn (F.T.); feiyangyan@hust.edu.cn (F.Y.); zoezhongys@gmail.com (Y.Z.); jinqian_li@brown.edu (J.L.); huigong@mail.hust.edu.cn (H.G.); 2Key Laboratory of Biomedical Engineering of Hainan Province, School of Biomedical Engineering, Hainan University, Haikou 570228, China

**Keywords:** brain–computer interface, electrode, multimodal, optogenetic

## Abstract

The brain–computer interface (BCI) is one of the most powerful tools in neuroscience and generally includes a recording system, a processor system, and a stimulation system. Optogenetics has the advantages of bidirectional regulation, high spatiotemporal resolution, and cell-specific regulation, which expands the application scenarios of BCIs. In recent years, optogenetic BCIs have become widely used in the lab with the development of materials and software. The systems were designed to be more integrated, lightweight, biocompatible, and power efficient, as were the wireless transmission and chip-level embedded BCIs. The software is also constantly improving, with better real-time performance and accuracy and lower power consumption. On the other hand, as a cutting-edge technology spanning multidisciplinary fields including molecular biology, neuroscience, material engineering, and information processing, optogenetic BCIs have great application potential in neural decoding, enhancing brain function, and treating neural diseases. Here, we review the development and application of optogenetic BCIs. In the future, combined with other functional imaging techniques such as near-infrared spectroscopy (fNIRS) and functional magnetic resonance imaging (fMRI), optogenetic BCIs can modulate the function of specific circuits, facilitate neurological rehabilitation, assist perception, establish a brain-to-brain interface, and be applied in wider application scenarios.

## 1. Introduction

A BCI is a device that enables communication between the brain and a computer or external device [1]. With half a century of development since J. J. Vidal first established the EEG-based BCI in 1973 [2], BCIs have evolved into some of the most powerful tools in neuroscience. Brain–computer interfaces usually consist of three components: a signal acquisition system, a signal processing system, and a controllable external device. External stimulation has been introduced to BCIs as a form of feedback [3,4], and brain stimulation has been utilized as feedback modulation in some BCI systems to improve their performance. Brain stimulation refers to the use of various external methods to directly or indirectly stimulate specific regions or neural circuits of the brain to activate or inhibit specific brain functions. This is mainly performed to improve the brain’s ability to control BCIs [5,6,7] or to ameliorate specific neurological disorders and investigate related brain mechanisms [8,9,10,11]. Researchers have been devoted to exploring brain stimulation methods that are more user-friendly, precise, and functional to enhance the performance of bidirectional BCI systems involving brain stimulation.

Over the past century, numerous brain stimulation techniques have been developed and refined. These techniques can be broadly categorized into two main types: electromagnetic stimulation [12,13], ultrasound stimulation [14,15] and light stimulation [16]. Electromagnetic stimulation involves the direct or indirect use of electrical currents to modulate specific brain regions through electrodes or electromagnetic induction technology. Examples include deep brain stimulation (DBS), transcranial electrical stimulation (TES), and transcranial magnetic stimulation (TMS) [17]. Ultrasound stimulation uses sound waves to act on mechanically sensitive voltage-gated ion channels, which are opened by mechanical deformation, thereby modulating the brain. Ultrasound stimulation is mainly divided into Transcranial Focused Ultrasound (tFUS) and Transcranial Unfocused Ultrasound (tUS) [18]. Light stimulation involves the regulation of neurons and neural circuits by utilizing light-sensitive receptors expressed in target neurons of experimental animals, a technique known as optogenetics [19].

Electromagnetic stimulation is a commonly used brain stimulation method that offers advantages such as high efficiency, high temporal resolution, simplicity in experimental procedures, and suitability for human subjects. Therefore, it has been used more widely in BCIs, such as the abovementioned techniques [5,7,10,11]. However, it still has limitations in terms of spatial resolution and diversity of regulation [20]. Ultrasound stimulation is a relatively new stimulation technique that has been used in a few brain–computer interfaces [6]. It has the advantage of better penetration and higher spatial resolution [21]. However, due to the relatively short period of time since the emergence of this technique, it still needs more relevant research and therefore has not been widely utilized in brain–computer interfaces. Like ultrasound stimulation, Optogenetic stimulation is also a new technique compared to electromagnetic stimulation that utilizes genetic engineering techniques to selectively express light-sensitive ion channel proteins in target neurons, enabling precise modulation of specific types of neurons [19,22]. Furthermore, different light-sensitive ion channel proteins can be expressed in target neurons based on specific regulatory requirements, and they can be activated or inhibited by light stimuli of different wavelengths. For example, neuronal activation usually involves a non-specific light-sensitive cation channel protein, channelrhodopsin (ChR2) [23], which opens up when exposed to blue light at approximately 470 nm, allowing a large number of cations to flow into the cell, leading to cellular depolarization and then causing nerve cell excitation. The inhibition of neuronal excitation is usually dependent on halorhodopsin (NpHR), a light-sensitive chloride pump [24] that transports chloride ions into the cell when irradiated with yellow light at approximately 570 nm, thereby inhibiting excitation of neuronal cells. The mechanism of action of these two ion channels is shown in Figure 1a. Additionally, studies have demonstrated that the precise targeting of optogenetic probes to individual neurons using specialized techniques can achieve single-cell precision neuromodulation [25,26,27]. Although optogenetics has insufficient penetration depth and safety and the ethical issues associated with genetic engineering [28], it still has the following advantageous capabilities:Targeted regulation of specific kinds of cells or specific locations with high spatial resolution [29].The bidirectional regulation of neurons, i.e., activation or inhibition of neurons, and diversity of regulation [30].Single-neuron activity manipulation and, in combination with diverse modulation modes of excitation light, multi-scale modulation from cells, loops, and brain regions to the whole brain [31].

These unique advantages make optogenetic stimulation far more effective than electromagnetic stimulation in various brain stimulation applications. When combined with signal detection systems and processing software, optogenetics can be integrated into novel BCI systems.

Although the optogenetic BCI is a relatively new technique that has emerged in recent years, it has made rapid progress and is being widely employed in labs. Here, we discuss the development and innovation of different hardware devices and software modules used in optogenetic BCIs along with examples of their applications in various fields. Finally, we present a discussion and prospects regarding software and hardware development, application scenarios, and multimodal integration.

## 2. Development

In recent years, open-loop-controlled optogenetic BCIs have been adopted in numerous studies on neural mechanisms [32,33,34] (Figure 1b). These mechanisms are externally stimulated by optical devices (lasers, optical fibers, or micro-light-emitting diodes (μLEDs)) to deliver precise stimulation to the brain for the regulation of neural activity. In neuromodulation, open-loop stimulation is capable of changing various state indicators such as the electrophysiological signals of mice and can be used to conduct experiments such as research on neural mechanisms and intervention and induction of neurological diseases. However, open-loop stimulation systems rely on manual regulation of stimulation parameters and timing, requiring trained operators and considerable work. Consequently, closed-loop BCI systems have emerged. These systems are capable of real-time analysis of the brain state and providing instantaneous feedback to control the output device [35,36,37]. Such closed-loop systems incorporate a feedback loop where a portion of the neural output is fed back into the system. Together with the desired output, this feedback mechanism regulates the input of the system to ensure that the system output closely aligns with the expected output and remains stable.

In closed-loop system (Figure 1b), a recording system is necessary for reading biological signals in various techniques like electroencephalogram (EEG), functional near infrared spectroscopy (fNIRS) [38], magnetoencephalography (MEG) [39], functional magnetic resonance imaging (fMRI) [40], functional transcranial Doppler ultrasonography [41], or photoacoustic fiberscope [42]. Recording systems consist of a sensor and supporting pre-amplification, filtering, and denoising circuits. Data processors, commonly in the form of computers or single-chip microcomputers and equipped with data analysis algorithms, are utilized to analyze the recorded signals. Communication elements are employed to facilitate interconnectivity among these hardware components, predominantly relying on wireless technologies such as data lines or Bluetooth (Figure 1c). Throughout the development of BCIs, both the hardware devices and software modules have undergone numerous updates and optimizations, leading to a diverse range of advancements, partly summarized in Table 1. In the subsequent sections, we present a comprehensive overview of the significant and universal developments pertaining to each component. Additionally, we summarize key system properties in Table 1 to provide a concise summary of the overall system characteristics.

### 2.1. Recording System

A recording system is designed for the capture and preliminary processing of the bio-electricity signals in a neural system. In animal experiments, the most commonly used recording devices are those for brain activities, which features electrodes as its primary components along with preliminary signal processing equipment such as preamplifiers and band-pass filters. Electrodes, a crucial element of this system, can be categorized into two types: invasive and non-invasive. Invasive electrodes necessitate surgical implantation within the cranium or even deep penetration into the brain tissue. While this approach carries the risk of mechanical damage and infection in the surrounding brain tissue, it yields a high signal-to-noise ratio and exceptional spatial resolution. Typically, invasive electrodes are applied in experimental animals like mice. On the other hand, non-invasive electrodes acquire data from external locations on the scalp, thereby minimizing potential harm and complications. However, the signals obtained via non-invasive electrodes exhibit a lower signal-to-noise ratio and poorer spatial resolution, and the electrodes are susceptible to detachment or interference from external signals. These electrodes are primarily employed in human subjects.

Early invasive electrodes were typically wires with a diameter of hundreds of microns made of metal materials such as tungsten, and only a few independent electrodes were integrated into a recording system [51]. This electrode is low-cost and meets the requirements for recording electrical signals. It is necessary to surgically penetrate the skull or penetrate deep into the brain and fix this electrode to the skull with cranial nails. During electrode implantation, brain tissue damage, which can even lead to the death of experimental animals, and local inflammation or infection can easily occur after implantation, so the survival time of experimental animals after implantation is often only a few weeks. Because the invasive electrode penetrates the skull and directly contacts the cerebral cortex and even deeper into the brain, the signal is mainly recorded from a small piece of brain tissue around the electrode with relatively high spatial resolution and signal quality. Non-invasive electrodes are mainly used in humans, are large in size (refer to the electrodes used in ECG measurement), and are divided into dry, wet, and semi-dry electrodes according to whether conductive paste is required. Dry electrodes do not use conductive paste. Examples of dry electrodes include needle electrodes, with sharp tips penetrating the stratum corneum of the skin, and capacitive electrodes, designed using the principle of capacitance. When a wet electrode is employed, skin resistance is reduced via the application of a conductive paste to the skin of the site to be measured [52]. The dry electrode is greatly affected by skin resistance and limb movement, and the signal quality is relatively poor. The signal quality of the wet electrode is better than that of the dry electrode, but before applying conductive ointment, it is sometimes required to remove part of the hair and thin the stratum corneum, which causes damage to the skin. The conductive paste will also become dry due to evaporation, resulting in a decrease in conductivity, which is not conducive to long-term signal recording. The semi-dry electrode is a promising new type of electrode that combines the advantages of the easy use of the dry electrode and the higher signal quality of the wet electrode. The signal recorded by the non-invasive electrode is jointly generated by a large brain region under the electrode and is also shunted by the skull with poor conductivity, so the spatial resolution is low, the signal quality is poor, and multiple electrodes are required to improve accuracy and signal quality [53]. For example, 21 electrodes are used in a human EEG measurement with the 10–20 international standard lead system, while some more accurate high-density EEG electrode arrays use as many as 345 electrodes [54].

With the advent of microfabrication technology and flexible material technology, there have been notable improvements in the miniaturization, density, and biocompatibility of certain electrodes. Microelectrode arrays (MEAs) have been produced with electrode diameters as small as a few microns with adjacent electrode spacing in the tens of microns, effectively functioning as preamplifiers [55,56,57,58]. As a result of these advancements, the entire recording system has become lighter and smaller, enabling implantation beneath the skull and attachment to the surface of the brain. This approach mitigates the risk of infection and minimizes potential damage to brain tissue caused by electrode activity. In certain experiments, mice implanted with such electrodes have survived for periods exceeding one year. However, due to cost and processing challenges, microelectrodes have not yet completely displaced traditional electrodes. Traditional metal electrodes are still widely used in short-term animal experiments that do not require high prognosis standards.

Non-invasive electrodes offer an alternative solution, and semi-dry electrodes combine the convenience of dry electrodes with the high conductivity of wet electrodes [59]. Semi-dry electrodes are typically constructed using flexible polymer materials or nano-metal porous materials. The use of the conductive paste is accurately regulated by a microprocessor, with each electrode consuming only tens of microliters of conductive paste per use. This approach effectively enhances conductivity while leaving minimal residue on the skin, thereby addressing many of the challenges associated with high skin impedance observed with dry electrodes and the use of conductive paste in wet electrodes. The advancements in electrode technology have resulted in more accurate and convenient EEG measurements, enabling the utilization of BCI technology reliant on EEG measurements in human subjects. Consequently, the quality of life for certain individuals has been improved.

### 2.2. A Processor System

The processor system is responsible for conducting time–frequency analysis or other data processing tasks on the information acquired by the recording system. Its primary function is to extract pertinent features regarding the state of brain activity and administer specific patterns of stimulation. Typically, the processor system relies on computer-based architectures or embedded systems. Computers are suitable for EEG analysis in laboratories and hospitals, while embedded systems based on microprocessors make the processor system portable and implantable [44].

The advancement in microelectronics has led to significant improvements in the performance and miniaturization of microprocessors, enabling their integration with microelectrodes on a shared flexible material substrate or their independent placement within portable devices. However, since the microprocessor itself is primarily dedicated to computational functions, it is imperative to address the challenges associated with data storage and power supply. In this regard, peripheral processor systems offer ample space for accommodating solutions to these issues. These systems can readily incorporate storage devices such as conventional SD cards, as well as ordinary lithium batteries and other power supply devices, which provide higher performance at lower costs. In this method, the data lines can be used to facilitate data exchange with other devices and enable battery charging [49].

Processor systems integrated into flexible materials necessitate careful consideration of space and performance constraints, including the utilization of micro-SD cards, micro-batteries, and other micro-components. In systems worn on the body surface, charging and data transmission can still be accomplished through wired connections. However, wireless solutions must be devised for systems implanted within the body to address these functionalities. Presently, micro-Bluetooth communication modules are commonly employed for data transmission between the recording or stimulation systems implanted in the body and the processor system, as well as between the integrated BCI system within the body and external systems. This approach enables near real-time data transmission with minimal latency (approximately 22 μs). Regarding power supply, there are currently two primary methods being explored. The first involves the utilization of flexible materials or piezoelectric ceramics to convert the mechanical deformation of biological tissues resulting from motion into electrical energy [50]. The second method entails the integration of magnetic induction coils within the system, enabling wireless charging through electromagnetic induction [48]. Piezoelectric ceramics, used for power supply, exhibit a significant piezoelectric constant, enabling efficient conversion of mechanical energy into electrical energy without the need for external power sources. However, larger ceramics are necessary to generate sufficient voltage (in the experiment, the piezoelectric ceramic size was 5 × 3 × 0.3 cm). Consequently, piezoelectric ceramics are more suitable for human equipment, such as adhering to limb joints or embedding within subcutaneous tissue. The power output of piezoelectric ceramics is dependent on the load resistance and is typically approximately 1 mW at a 100 kΩ load [50]. On the other hand, miniature magnetic induction coils integrated into flexible materials are fabricated using photolithography techniques, limiting the manufacturing of multilayer coils. Instead, multi-turn wires are printed on the same plane to increase the number of turns. However, the power conversion efficiency of these coils still requires further improvement. Powering devices within the body using magnetic induction coils still relies on external devices, but the advantage of these coils lies in their smaller size, allowing for implantation in any body location. They are particularly suitable for placement within static tissues like internal organs or intracranial tissues. Furthermore, numerous research groups are actively working on reducing the power consumption of various system components, aiming to extend the operating time achievable with a single charge [60]. These efforts enable the long-term operation of implant systems, including sensor and stimulation devices, within the body, thereby facilitating their regulatory and therapeutic applications.

The algorithm plays a crucial role in determining the information derived from the acquired signals, analyzing this information, and generating corresponding signals to control the behavior of the stimulation system. For instance, in systems related to prosthetic design and development, acquiring information about direction and speed is essential. Early approaches involved the utilization of the population vector algorithm (PVA) [61] and optimal linear estimator (OLE) for fitting followed by the implementation of Kalman filters [62]. In systems focusing on the detection and treatment of neurological diseases, the emphasis lies on the time–frequency characteristics of EEG signals. Consequently, a variety of time–frequency analysis algorithms are commonly employed. Fourier transform (FT) and Fast Fourier transform (FFT) are widely utilized due to their simplicity and reliability, particularly in disciplines involving stationary periodic signals. However, some algorithms are more suitable for analyzing non-stationary and non-periodic biological signals like those of an EEG. Examples of such algorithms include the Choi–Williams distribution (CWD) algorithm proposed by H. Choi and W. Williams [49], the Hibert–Huang Transformation (HHT) algorithm proposed by Hibert and Huang [50], and the wavelet transform (WT) method. These algorithms have been widely applied in extracting time–frequency information from biological signals.

BCIs designed for various purposes require tailored utilization of this information, leading to the development of customized algorithms. The optimization of algorithms primarily focuses on the solution of problems like the end effects and over/undershoots of the above time–frequency analysis algorithms [63], as well as reducing the complexity of the algorithms to improve their computational efficiency, reduce latency, and minimize power consumption. For instance, Junwen Luo et al. successfully achieved continuous recognition and regulation of specific EEG patterns in mice using the BCI toolbox algorithm based on wavelet transform and a proportional–integral–derivative (PID) algorithm. This implementation demonstrated low power consumption (9.7 mW) and low latency (22 μs) [49].

Deep learning techniques outperform support vector machines (SVMs) in terms of classification accuracy. An example of such an advancement is the MC-SleepNet algorithm, introduced by Masato Yamabe et al. This algorithm integrates convolutional neural networks (CNNs) and long short-term memory (LSTM) artificial neural networks to discriminate the sleep state of mice. Remarkably, it achieves an impressive accuracy of 96.6%, surpassing the accuracy rate of algorithms that do not incorporate neural networks, which typically reach approximately 80%. Additionally, the algorithm demonstrates improved processing speed, further enhancing its efficiency [64].

### 2.3. Stimulus System

The stimulus system refers to a system that applies precise patterns of stimulation to specific brain regions to induce targeted neural activities. In optogenetic BCIs, the primary sources of stimulation include lasers, optical fibers, and μLEDs. In the experimental design, optogenetic stimulations to induce specific activities through optogenetic methods not only require an appropriate wavelength of light but also the selection of appropriate stimulation frequencies and power to follow the light attenuation law and safe irradiation [65]. The aim of this approach is to effectively activate neurons in the target area while minimizing heat damage to brain tissue. The most commonly used optogenetic protein, ChR2, requires 470 nm blue light activation. Before the advent of blue light-emitting diodes (LEDs), only larger laser sources could produce blue lasers. These laser sources are not easy to move, and the resulting laser light needs to be transmitted by a fiber and projected into a designated brain region by a short, opaque, black, ceramic fiber that penetrates the skull. This method has been extensively used in animal experiments due to the consumable properties of this fiber. However, it exhibits significant limitations in terms of free mobility, precise stimulation targeting, potential damage to brain tissue, and a heightened risk of infection. Some teams have taken inspiration from microelectrode arrays, such as those based on chemical gas phase deposition, to develop micro-optical arrays capable of guiding light. This approach enhances the spatial resolution of photostimulation and reduces the mechanical damage caused by optical fibers to brain tissue [66]. Nevertheless, this approach still necessitates the use of an external optical fiber penetrating the skull to deliver the laser source from the external environment. As a result, it does not fully mitigate the risk of infection associated with traditional photostimulation or the potential postoperative damage resulting from the mechanical movement of the optical fiber.

The widespread availability of blue LEDs has significantly expanded the potential applications of μLEDs in the field of optogenetics. μLEDs possess inherent characteristics such as compact size and low power consumption [67]. These features enable their integration with flexible materials and direct placement in proximity to neurons requiring stimulation, thereby greatly enhancing stimulation-targeting precision. Moreover, μLEDs can be directly integrated into microelectrodes, enabling the creation of composite flexible electrodes or electrode arrays capable of both collecting electrical signals and delivering light stimulation. This integration of composite electrode arrays with microprocessor systems forms fully implantable BCI systems, enhancing the security, reliability, and durability of BCI technology within the body.

Another development trend in stimulation modules is to achieve single-cell stimulation through light modulation. In recent years, two types of light-targeting strategies have emerged for single-cell photostimulation, commonly known as serial scanning and parallel-patterned light targeting [68]. Serial scanning enables single-neuron precision of optical stimulation by focusing the excitation beam on a small, diffraction-limited spot. To expand the stimulation target, a series of predefined regions of interest (ROIs) can be targeted, moving the light spot laterally within the excitation field. Scanning the beam does not change the spot size, so the type of scanning unit does not affect the spatial resolution. The spatial positioning accuracy and flexibility of scan-based optical stimulation relies on the ability to deflect the illumination beam at precise angles and scan arbitrarily defined trajectories across the sample. In order to reduce the scanning time, a discontinuous scanning mechanism through the acousto-optical deflector (AOD) vector mode method has been the subject of research. This approach is combined with a galvanometer (GM) to achieve the fastest calculation speed scan mode to access a user-defined set of ROIs [69].

The parallel technique involves coupling the microscope objective to a spatial light modulator (SLM), which acts on the phase or intensity of the incident light beam. Intensity modulation technology mainly obtains illumination patterns by directly shaping the spatial intensity of light in a plane conjugated to the sample plane, including micro-light emitting diode (micro-LED) arrays and extended light sources for uniform illumination [70]. Light shaping is obtained by turning on/off each emitter of the array independently. Each pixel of the SLM [71] independently modulates the local light intensity by extending the amplitude of the light source’s uniform illumination. The amplitude SLM strategy can be divided into two types: a digital micromirror device (DMD) [72] and liquid crystal SLM. Phase modulation technology mainly includes computational holograms (CGHs) [73]. It is a wavefront modulation technique that can shape laser light into different shapes, including 3D distributions of diffraction-limited points and arbitrarily extended patterns. In addition, the above technology combined with two-photon stimulation can increase the depth of regulation and ensure *z*-axis resolution [74].

Taking into account all these imaging and stimulation options, as well as the possibility of combining single-photon imaging and two-photon optical stimulation, when constructing cellular or near-cellular optogenetic control systems, we need to consider many factors, especially field of view, spatial resolution, and expected trade-offs between temporal resolution. Despite recent developments in compressed sensing and the improved use of prior information on sample structure [72,75,76], basic spatiotemporal constraints still exist.

## 3. Applications

An open-loop stimulation strategy for commercialized equipment has predominantly been employed in previous studies. This strategy offers convenience to researchers in designing stimulation parameters and timing according to experimental requirements. In the field of neuromodulation, open-loop stimulation has proven sufficient for altering state indicators such as electrophysiological signals, enabling research on neural mechanisms, intervention and induction of neurological diseases. In these experiments, light stimulation parameters (e.g., pulse frequency) were chosen based on information from the literature or direct neural recordings rather than direct observation and feedback on the neural effects [77,78]. Despite this, the relationship between neural activity and behavioral states at various scales was still obtained in these studies, and these techniques are widely used in the study of mechanisms of locomotion [32,33], learning and memory [34,79,80], sleep [81,82], emotion [32,83,84], pain [82,84,85], and other functions. In these studies, the function of the target neural circuit is usually revealed through statistical analysis comparing behavioral performance before and after optogenetic modulation of the circuit or whether or not optogenetic modulation is implemented. Moreover, when combined with electrode recording and other technologies, it can make up for the problem that the recording results are not neuron-specific. Using optogenetic tools to activate specific types of neurons and recording with electrodes, the discharge of target neurons can be distinguished based on the strong synchrony between action potentials and laser firing times [86].

However, due to the complexity of the nervous system, some mechanism exploration experiments and functional enhancement or restoration experiments place high demands on the firing time and firing pattern of external perturbations. In addition, open-loop regulation may not meet expectations or may cause too many side effects. Therefore, closed-loop systems that record and analyze neural signals in real time to regulate stimulation parameters have become a research hotspot in recent years. To fully exploit the potential of closed-loop optogenetics, optical interventions must continue to more closely resemble natural loop activity dynamics in scale and complexity. Closed-loop optogenetics uses simultaneous readings of neural activity or behavior to decide in real time how and when to deliver optogenetic stimulation, using the measurements to guide stimulation in a closed feedback loop.

Closed-loop control can be used to establish loop-level feedback control [87]. Neurons communicate by generating trains of action potentials, encoding the information into mean firing rates or action potential firing times. Therefore, by designing sequences of light pulses, the firing pattern of any particular neuron can be simulated. Based on this, the effects of the generated firing patterns on postsynaptic neurons or corresponding behavioral phenotypes can be studied [88]. For example, Sohal et al. used real-time feedback closed-loop firing of optogenetic stimuli that simulated pulse triggering of pyramidal neurons to drive inhibitory parvalbumin-positive interneurons and induce gamma oscillations [89]. Furthermore, closed-loop stimulation showed that the gamma has a causal effect on the efficient flow of information in the circuit, whereas randomly removing the same number of spikes does not have this effect [89]. Similar effects are also seen in stimulus temporal encoding. Optogenetic stimulation was employed in another study to suppress neural activity in the dorsal hippocampal CA1 region during specific phases of the endogenous theta rhythm in freely moving mice [90]. By combining behavioral experiments involving spatial navigation in an “I”-shaped maze, the researchers observed that delivering inhibitory stimuli at the peaks of theta waves improved navigational accuracy and reduced the time spent on incorrect exploration when spatial cues indicating the location of rewards were provided during the encoding phase of the task. Conversely, issuing inhibitory stimuli at the troughs of theta waves during the retrieval phase increased the likelihood of the mice making correct choices at intersections based solely on their decision-making (Figure 2).

The optogenetic paradigm provides many new ideas for medical applications. Nerve-related diseases and pain affect the lives of many people. Research on neuromodulatory systems focuses on intervention and the treatment of these diseases. The research reports that optogenetic stimulation can treat diseases by bidirectionally regulating neuronal activity [85,91] and enhancing neuronal plasticity [92,93]). In 2021, Fakhoury M et al. conducted a study on depression and depression-like behaviors utilizing open-loop optogenetic stimulation devices [94]. The objective was to alleviate depressive symptoms through neural regulation. This comprehensive investigation, involving multiple brain regions with a particular emphasis on the medial prefrontal cortex (mPFC) and ventral tegmental area (VTA), revealed the complex neural circuitry underlying depression. The study highlighted the involvement of various brain regions, cell types, neurotransmitter systems, and neural pathways in the manifestation of depressive symptoms. Additionally, the study identified multiple stimulation sites capable of inhibiting depressive-like behaviors, offering a potential avenue for the treatment of depressive symptoms. Furthermore, numerous studies have explored the structural and functional interventions of neural circuits involved in memory formation [95], functional impairment [96], epileptic seizures [97], and chronic pain generation [98]. These investigations provide valuable insights into the underlying mechanisms of these conditions and offer potential strategies for therapeutic interventions.

Likewise, the application of closed-loop systems in this field is also a research hotspot. For instance, in 2015, Ramin Pashaie et al. developed a closed-loop system that combines micro-electrocorticography (micro-ECoG) and fluorescence imaging to acquire input signals and deliver modulated excitation light output via a DMD [99]. This system was successfully employed for stroke detection and regulation in non-human primates. Fluorescence imaging was utilized to monitor blood flow, and the output devices stimulated vascular-coupled neurons to modulate vasodilation when substantial reductions in blood flow in major cerebral vessels were detected. Similarly, Bing-Hong Lin et al. extracted sample entropy and frequency band information from contralateral depth electroencephalography signals in mice, enabling the classification of acute epilepsy and chronic epilepsy with a detection accuracy exceeding 90%. Additionally, employing proportional-plus-off control to manage the optogenetic device for delivering stimulation effectively suppressed epileptic seizures. The average success rate of epilepsy suppression ranged from 86% to 98% during the acute phase and reached 94.3% during the chronic phase [8] (Figure 2). In 2020, researchers developed a lightweight, miniaturized, low-power, head-mounted, closed-loop, optogenetic stimulation device capable of distinguishing normal and abnormal neural signals. When abnormal neural activity is detected, the device emits laser light for stimulation [95]. More recently, Qiaosheng Zhang et al. recorded real-time electrical signals from the anterior cingulate cortex of rats to detect pain onset and utilized optogenetic stimulation to activate the prefrontal cortex during pain onset. This intervention effectively inhibited acute mechanical pain, thermal pain, chronic inflammation, as well as sensory and affective aspects associated with neuropathic pain [100].

Another development direction of closed-loop control systems is that prostheses capable of virtual sensory feedback and neural output decoding can provide users with the experience of real limbs. In 2018, successful virtual touch perception in mice was achieved in a study through the utilization of optogenetic stimulation. Aamir Abbasi et al. employed an internal imaging system to precisely locate the brain region within the primary motor cortex (vM1) corresponding to each whisker (barrel column). By controlling the firing rate of vM1 within a specific range, the mice were able to control the movement of their specific whisker. When the mouse controls its specific whisker movement, light stimulus feedback is given in the primary somatosensory cortex (vS1), as well as the reward of water. The mice had to lick at the same time to obtain the reward. In the study, it was proved that the licking action of mice was dependent on the artificial light stimulation feedback provided in vS1, and it was proved that the light stimulation made the mice feel that they were in contact with the virtual stick that provided water [101] (Figure 2).

In addition, some research focuses on brain–brain interfaces and explores ways to achieve mind control and telepathy. A groundbreaking study introduced the application of optogenetics to construct a BCI in mice. The system captures calcium imaging signals associated with movement velocity within the brainstem nucleus (NI) of freely moving mice. These signals are utilized to encode optogenetic stimulation in controlled mice, thereby regulating their movement patterns to closely resemble those of the source mice. In this study, the high-speed transmission of fluorescent signals employed in optogenetics plays a crucial role in rapidly conveying neural information and enabling real-time, synchronized control of ideation [102] (Figure 2).

**Figure 2 bioengineering-11-00821-f002:**
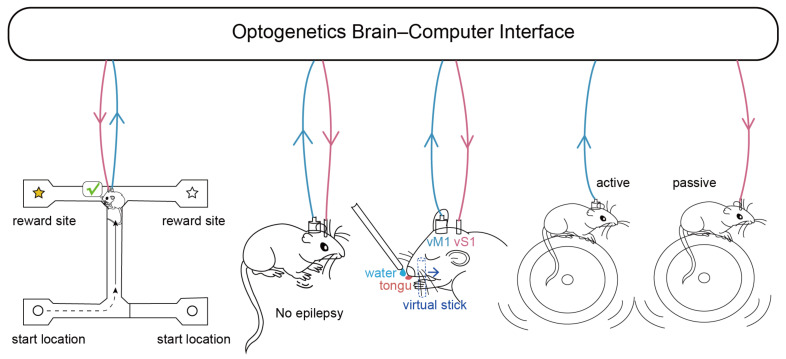
Examples of optogenetic BCI applications. In an example of enhancing cognitive function, optogenetic stimulation of firing inhibition on the troughs of endogenous theta rhythms in the mouse hippocampus increased the probability of correct selection when the mice were in the retrieval arm [90]. In an example of neurotherapy, detecting the onset of epilepsy and providing optogenetic stimulation can be effective in alleviating it [8]. In an assisted perception example, the primary somatosensory cortex (vS1) was given light stimulus feedback along with a water reward, while the mouse controlled the primary motor cortex (vM1) firing rate within a certain range, that is, mouse-specific whisker movement [101]. After training, the mouse’s licking behavior relied on the feedback of artificial light stimulation provided in vS1, which proved that the light stimulation made the mice feel that they were in contact with the virtual stick that provided water. In a brain–computer–brain interface example, calcium imaging signals related to locomotion velocity in the brainstem nuclei (NI) of autonomously moving mice were used to encode optogenetic stimuli in controlled mice, modulating their locomotor patterns so that they closely mimicked the movements of the active locomotor mice [102]. Blue arrows indicate brain-machine information flow and pink arrows indicate machine-brain information flow.

## 4. Discussion and Prospects

The future development trajectory of optogenetic BCIs is a topic of significant interest for numerous research groups, given its immense potential for practical applications [103]. In order to provide a comprehensive perspective, this discussion encompasses three key directions: software and hardware advancements, application scenarios, and the integration of multimodal BCIs. These developmental prospects are formulated in response to diverse requirements, as illustrated in Figure 3a.

### 4.1. Software and Hardware

In research on optogenetic BCIs, the use of invasive electrodes is an unavoidable topic for researchers. These electrodes, implanted into the brain through the skull, enable the detection of bioelectrical signals, which serve as the gold standard for monitoring brain activity. However, conventional electrical interfaces, typically in the form of electrode probes or flexible electrodes [104,105,106], are characterized by their large size and high modulus [105]. This renders them incompatible with neural tissue [107]. The elastic modulus of neural tissue is generally less than 100 kPa. The mechanical mismatch between tradition-al electrode materials and soft neural tissue creates shear forces, leading to acute inflammatory responses during implantation, such as immune cell activation, migration, and local ischemia. After electrode implantation, chronic inflammatory responses can occur during tissue movement and electrode micromotion. Persistent inflammation in the damaged area can lead to reactive astrogliosis. Over time, layers of activated microglia in-filtrating macrophages, reactive astrocytes, and migrating meningeal fibroblasts can encapsulate the neural implant, ultimately increasing the impedance of the neural interface, isolating the electrode, reducing the signal-to-noise ratio, and causing the failure of the neural interface function [57,108]. Moreover, the limitations of traditional interfaces result in reduced signal resolution and potential damage to peripheral neural tissue when using optical fibers, in contrast to the high spatiotemporal resolution pursued by optogenetic BCIs. Currently, the coupling of waveguides and neural soft tissues using high-modulus glass optical fibers, as the prevailing photostimulation hardware, does not yield ideal results in animal experiments.

To address the aforementioned challenges, the prevailing approach in the field is to explore alternative solutions to traditional electrodes that offer enhanced tissue compatibility and flexibility or employ highly integrated microcomposite electrodes [109,110]. For instance, transparent conductors combined with photostimulation [111,112,113,114,115] or photostimuli–electrophysiological recording miniature arrays composed of silicon probes and titanium iridium electrodes [116] can be employed. Although these investigations are still in the early experimental stage, they provide valuable insights into the future direction of research in this field.

The wired connection between the brain and the photoelectric interface can interfere with the natural behavior of experimental animals. Consequently, the future hardware development direction for optogenetic BCIs is anticipated to be the wireless photoelectric combined BCI. This wireless interface is designed to achieve high tissue compatibility and integration. However, it also necessitates resolving the challenge of high power consumption in wireless sensors, as excessive heating can lead to thermal damage to surrounding tissues. In addition, with the elimination of cable connections, the wireless photoelectric combined BCI imposes higher requirements for the signal transmission rate, the signal processing rate, and the stability and accuracy of encoding and decoding within the system. Therefore, the software side will inevitably develop towards more efficient codec algorithms and more integrated software platforms.

### 4.2. Application Scenarios

BCIs possess significant potential for applications in the medical and health domains due to their ability to substitute traditional brain information input and output circuits. Specifically, they hold promise in the fields of circuit functional exploration, rehabilitation of limb disorders, monitoring and treatment of neurological diseases, and other related areas. While there are currently limited examples of optogenetic BCIs being employed in virtual sensory perception and brain-to-brain information exchange, the high spatiotemporal resolution and precision of optogenetic technology in regulating neural activity inherently make it highly suitable for such applications. As virtual sensation and induced neuroplasticity technologies continue to advance, the application of optogenetic BCIs in medicine is expected to progress in the following directions. First, assistive BCIs can aid patients with limb disabilities in controlling prostheses by transmitting signals related to motor intentions. Second, rehabilitative BCIs can facilitate the restoration of damaged neural circuits in the patient’s brain and enhance the recruitment of neuronal function.

Nevertheless, it is important to note that the application of optogenetics in humans is subject to stringent ethical constraints, as it typically involves the integration of genetically modified viruses into the host. Current research endeavors have focused on in vitro cultivation of human pluripotent stem cells (hiPSCs) to construct neural and muscular models. In these models, the expression of optogenetic tools in human cells has been investigated [117]. Additionally, related studies have utilized optogenetic regulation to establish various disease models, thereby facilitating further exploration of disease pathogenesis and potential treatment options [118].

### 4.3. Multimodal BCI

As optogenetic BCIs progress towards minimally invasive, lightweight, and versatile application scenarios, researchers are also exploring the integration of various technologies into their recording systems. These interfaces will likely evolve towards a multimodal approach in the future. A multimodal BCI refers to a hybrid system that combines different modalities, such as an EEG, functional near-infrared spectroscopy (fNIRS), functional magnetic resonance imaging (fMRI), magnetoencephalography (MEG), and functional ultrasound (fUS) [103,119,120]. By incorporating multiple modalities, researchers aim to enhance the accuracy, resolution, and functional characterization of brain activity measurements in BCIs.

In hybrid systems, fMRI and fUS are commonly employed as complementary techniques to the EEG and fNIRS methods for brain imaging, primarily due to their superior spatial resolution, which serves as the benchmark for evaluating these two BCI systems. During closed-loop, in vivo, electrophysiological recordings using optogenetics, scientists activate or inhibit neurons through optogenetic methods to study the correlation between neuronal electrical activity, electrophysiological changes in circuits, and behavioral outcomes. Combining fMRI technology with fiber photometry enables fMRI scans to achieve cellular resolution, validating the relationship between changes in astrocytic Ca^2+^ levels and BOLD signals and elucidating the contributions of individual neuronal subpopulations to brain neural circuits [121]. fUS is a recently developed technology that enables the recording and decoding of spatiotemporally precise activity patterns from large brain regions. fUS neuroimaging utilizes ultrafast pulse–echo imaging to simultaneously detect changes in cerebral blood volume (CBV) across multiple brain areas. These CBV changes are closely associated with single-neuron activity and local field potentials. The technique exhibits high sensitivity to slow blood flow (~1 mm/s) and offers excellent spatiotemporal resolution (100 μm; <1 s) with a large and deep field of view (~2 cm). This allows for the effective acquisition of high-resolution spatial information with minimal invasiveness, thereby enhancing the ability of invasive closed-loop systems to collect intracranial data [122]. On the other hand, MEG and the EEG offer high temporal resolution. However, their sensitivities to radial, tangential, extracellular, and intracellular currents still differ. By combining the EEG with EMG and fMRI or fUS, BCI systems can improve decoding performance [123].

An EEG based on neuronal electrical activity is used to detect brain activity through the recording of spontaneous and rhythmic neuronal potentials beneath the scalp surface. On the other hand, fNIRS detects hemodynamic changes through the neurovascular coupling mechanism [124,125]. Similarly to fMRI, functional near-infrared spectroscopy (fNIRS) leverages the excellent scattering properties of blood’s main components for near-infrared light in the 600–900 nm range, as well as the differential absorption characteristics of oxyhemoglobin (HbO_2_) and deoxyhemoglobin (Hb) at these wavelengths. Infrared light emitted from a source penetrates brain tissue and, after absorption and scattering, reaches a detector. The absorption and scattering processes adhere to the Beer–Lambert law. Using tomographic imaging algorithms, images can be reconstructed to indirectly detect changes in the concentrations of oxyhemoglobin, deoxyhemoglobin, and total hemoglobin within human tissue. This enables closed-loop systems to detect neural activity by exploiting the spectral absorption differences between oxyhemoglobin and deoxyhemoglobin [126]. In the current EEG-fNIRS hybrid detection system, most of the near-infrared probes and EEG electrodes are jointly fixed to the same probe cap [127]. The combination of the EEG and fNIRS techniques in hybrid systems offers several advantages, which have been elucidated in the literature.
(1)These techniques have a wide range of adaptations, from simple visual stimulation experiments to detection in intensive care patients, with an adaptation age from premature infants to elderly patients, covering all ages [128].(2)The two methods complement each other in terms of temporal and spatial resolution [129].(3)Both the EEG and fNIRS techniques have the advantages of being relatively small and inexpensive devices and can be integrated into portable devices [130].(4)The fNIRS and EEG techniques are robust to motion artifacts without excessive physical constraints.(5)Compared to other techniques, non-invasive EEG and fNIRS methods can be performed under conditions close to daily life, providing considerable freedom in experimental design.(6)The methods are silent, making them more conducive to language research and auditory cognitive experiments.(7)Integration is relatively simple due to the absence of electro-optic interference [109].

Numerous challenges are associated with multimodal BCIs. Firstly, it is essential to establish multi-scale modeling and analysis methods. The application of multimodal BCI closed-loop systems requires the development of cross-modal neural activity control schemes and the analysis of synchronously recorded multimodal neural activities. This approach is crucial for mapping neuronal activities across different scales to thoughts and behaviors within the same time frame. Secondly, there is a lack of established frameworks for analyzing high-dimensional datasets operating at different spatial and temporal scales. For instance, datasets emerging from multimodal optical and electrophysiological mapping of neuronal activities are scarce, and the collection of high-frequency, multi-channel, and multimodal data over long-term studies exacerbates issues related to data collection bandwidth and storage [131]. Lastly, the absence of a standardized experimental design and reporting reduces the comparability and reproducibility of datasets. Therefore, creating robust experimental platforms capable of connecting data in a standardized format is of paramount importance [132].

## Figures and Tables

**Figure 1 bioengineering-11-00821-f001:**
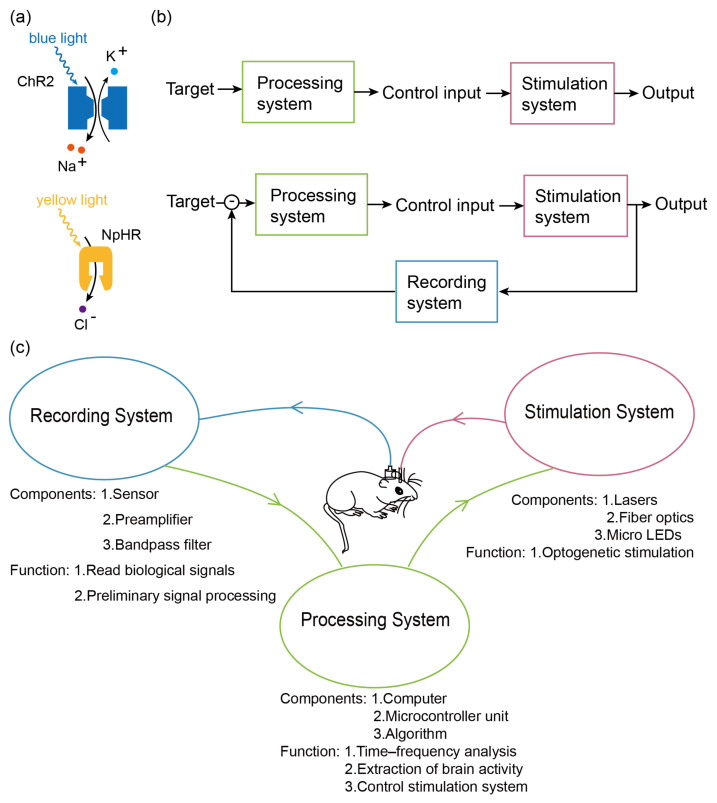
(**a**) Upper half: mechanism of ChR2. When irradiated with blue light, ion channels open, leading to sodium inward flow, causing depolarization. Lower half: NpHR mechanism. When irradiated with yellow light, the ion channel opens, leading to chloride ion inward flow, resulting in hyperpolarization. (**b**) Upper half: open-loop system. The output is generated directly through the processing and stimulation systems with no feedback control. Lower half: closed-loop system. Outputs are generated through the processing system, the stimulation system, and the recording system, using the recording system as a feedback control to modulate the outputs. (**c**) Frame diagram of the EEG-based optogenetic BCI. The recording system reads signals from the animal’s brain through electrodes, performs a series of pre-processing, and then transfers the data to the processing system. The processing system analyzes and decodes the signal read by the recording system and encodes the signal according to the analysis results. After the encoding is completed, the stimulus system is controlled to output the signal. The stimulus system gives corresponding optogenetic stimulation to the animal according to the encoding of the processing system.

**Figure 3 bioengineering-11-00821-f003:**
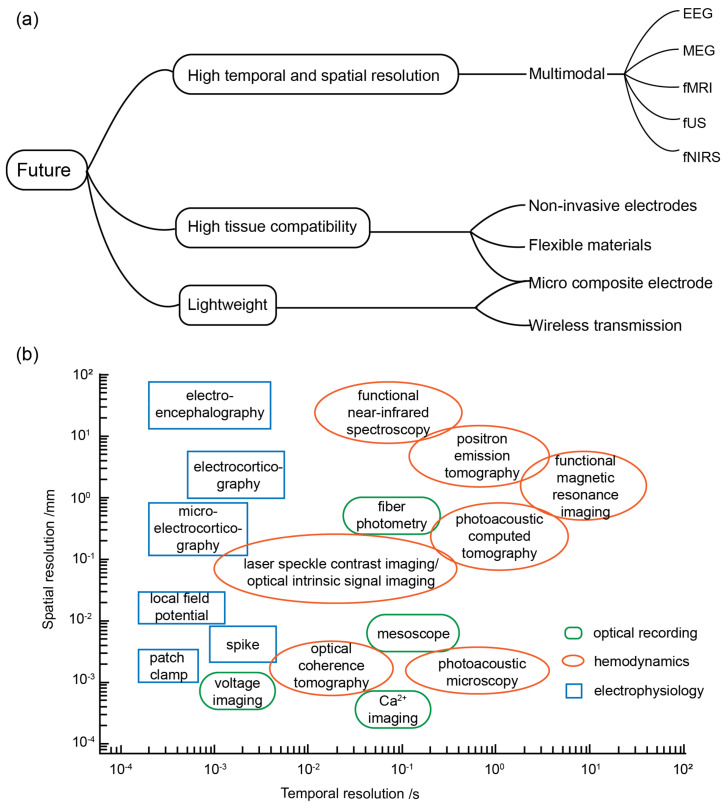
(**a**) Future development trends. (**b**) Spatial and temporal resolutions of different neural interface technologies.

**Table 1 bioengineering-11-00821-t001:** Representative BCI systems of different times.

Reference	Parts	Target	Advantages
Alizadeh-Taheri et al. (1996) [43]	Electrode only	Human	Non-invasive; Good electrical performance; No conductive paste.
Bertram et al. (1997) [44]	Recording system	Mice	Video signals combined with electrical signals; Suitable for long-term experiments.
Weiergräber et al. (2005) [45]	Recording system	Mice	Wireless data transmission; Suitable for long-term experiments.
Wu et al. (2008) [46]	Recording system	Mice	Suitable for young mice; High durability and low cost; Suitable for long-term experiments.
Etholm et al. (2010) [47]	Recording system	Mice	Wireless data transmission; High sampling rate; Light weight; Reusable device.
Mickle et al. (2019) [48]	The whole system	Mice	Fully implantable; High integration capacity; Flexible; Wireless data transmission and charging.
Luo et al. (2020) [49]	The whole system	Mice, human	Partially implantable; Reprogrammable; Wireless data transmission; Low transmission delay; Low power consumption.
Yang et al. (2022) [50]	The whole system	Mice, human	Non-invasive; High integration capacity; Flexible; Wireless data transmission; Self-powered.

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
