# Peer review of "Optogenetic Brain–Computer Interfaces"

_bioengineering, 2024, doi:10.3390/bioengineering11080821_

Round 1
Reviewer 1 Report
Comments and Suggestions for Authors
(line 33) – The authors hold the view brain stimulation is essential to BCI, and by extension that BCI's are by definition bidirectional. This should be nuanced, as within the community, a BCI is taken to be potentially bidirectional. Granted that a form of brain stimulation usually exists if one includes sensory stimuli, this does not appear the intended meaning. - To improve the introduction, the authors may highlight the benefits bidirectionality in BCI applications. The authors can also consider abandoning the introduction of BCI and focusing on brain stimulation as such, while mentioning BCI as an important application. Furthermore, it would be helpful if “brain stimulation” was given a more precise definition. Does it include sensory stimuli? I suspect it doesn’t, but in the context of BCIs, this should be clarified.
(line 35) “It not only facilitates feedback control of external BCI equipment, …” – This claim should be substantiated with a reference. For which BCIs has it been shown that brain stimulation facilitates controlling an external device?
(line 52) “suitability for human subjects.” – Clarify why optogenetics are not suited for humans. Or consider rephrasing this in terms of the risks associated with surgical procedures.
(line 75) “numerous studies on neural mechanisms” – As numerous studies exist, a few references to key publications on this topic could be relevant.
(line 80) “excessive consumption of artificial resources.” – Clarify “artificial resources”.
(line 88) “In a closed-loop system, in addition to the stimulation devices described above, there are recording systems for reading biological signals” – The wording is a bit confusing.
(Figure 1) – MCU does not need be abbreviated ~ Microcontroller unit (Figure 1) – Stimulus system ~ Stimulation system ~ Stimulator
(line 114) – “The recording system is a sophisticated apparatus …” – Try to be concise ~ “The recording system captures and analyzes brain activity.” Furthermore, Figure 1 indicates the recording system preprocesses the captured brain signal, but the processing system analyzes the signals.
(line 119) – EEG (electroencephalography) is a non-invasive recording technique. Other, closely related techniques, such as stereo-EEG and electrocorticography are invasive.
(line 120) – “even penetrating deep into the brain tissue.” – Again, other recording techniques (e.g. microelectrode arrays or single cell recordings). Consider talking about signal types rather than recording techniques (e.g. local field potentials, spikes), as the remainder seems to focus on invasive recordings.
(line 146) “conductive paste containing chloride ions” – Many gel-based solutions exist, even salt and chloride-free ones.
(line 150) “it is necessary to remove part of the hair” – Removal of hair is extremely rare in human EEG experiments and unnecessary for most applications.
(line 155) “but its appearance is late” – Improve the wording.
(line 189) – It would seem the stimulus system administers the stimulation, while the processing system controls (or computes) the amount of stimulation to be delivered. Note that while time-frequency analysis is common, it is not essential to processing brain signals.
(line 193) “Computers have historically …” – It is sufficient to mention the miniaturization of micro-processors, as they are the heart of every computer. (line 204) – Indeed, portability also benefits from miniaturization of the data storage (and transmission) and power supply.
(line 245) “The algorithm plays a crucial role …” – Improve the wording.
(line 254) “The Fourier transform (FT) and its improved variant, the Fast Fourier transform” – FFT is not an improvement of variant of FT, rather it used for discrete io. continuous signals. However, it is more speed and memory efficient than the normal DFT (Discrete Fourier Transform).
(line 274) “However, the application of deep learning techniques has significantly enhanced the operational efficiency and accuracy of this type of algorithm” – Improve the wording ~ “DNNs outperform SVMs in terms of classification accuracy”
(line 288) “appropriate wavelength of light but also the selection of appropriate frequencies” - Clarify the difference between selecting the wavelength and the frequency of light.
(line 290) – Clarify what causes the damage to brain tissue, e.g. heat.
(line 300) “have explored the implementation of microelectrode arrays” – Improve the wording ~ “have taken inspiration from microelectrode arrays, …”
(line 356) “The closed-loop optogenetic system offers several advantages, including neuron specificity, high spatiotemporal precision, and efficient stimulation, …” – Redundant (repeated) information on closed-loop system.
(line 359) “in a study” – Provide reference.
(line 406) and elsewhere – Provide references (e.g. [56]) at the start of the paragraph discussing the referenced work.
(line 461) “In research on optogenetic BCIs, the use of invasive electrodes is an unavoidable topic for researchers. These electrodes, implanted into the brain through the skull, enable the detection of bioelectrical signals, …” – Improve wording, and avoid redundancy and wordiness.
(line 465) “… flexible electrodes … characterized by their … high modulus. This renders them incompatible with neural tissue.” – Clarify how low a modulus should be to be compatible with brain tissue.
(line 526) “In hybrid systems, fMRI and fUS…” – Clarify how these hybrid systems relate to optogenetic BCIs. How will they improve the closed-loop system?
(line 537) “The combination of EEG and fNIRS in hybrid systems” – Clarify how these hybrid systems relate to optogenetic BCIs. How will they improve the closed-loop system?
(General)
The main goal of the work is to situate optogenetics with the field of BCI and to review the technical advances within optogenetics. This ambitious attempt resulted in a low-quality overview of BCIs.
Overall, the work would benefit from a rigorous restructuring and from taking inspiration from previous BCI review papers (e.g. [ref]). This would allow to focus on the history, advances and practical applications of optogenetic BCIs in particular. However, the work would still lack a clear contribution and novelty.
The work covers many interdisciplinary aspects of BCI design. While the presented information is interesting, the discussions often lack relevance, precision and/or completeness. As a review paper, the work lacks coherence and a conceptual framework. A few works by other authors using optogenetic stimulation are discussed, without distilling any novel insight.
References
(line 555) can be removed.
Comments on the Quality of English Language
The quality of English is below standards
Author Response
请参阅附件。

Reviewer 2 Report
Comments and Suggestions for Authors
The authors develop an optogenetic brain-computer interface. It is interesting for readers. However, some things could be improved, and these are listed below.
1. I suggest that the authors need to rewrite the abstract section. The motivation, proposed, and experimental results should be detailed.
2. In the introduction section, the authors should detail the major contribution of this study. What problems are solved?
3. In the development section, many literal reviews or system comparison should be move to the introduction section. The authors should systematically review the related systems.
4. The applications section is unsuitable description. If it describe that the proposed system can be used in these applications, it should be move to the introduction section. In this section, the authors need to detail the system performance of the proposed approach.
5. Finally, a conclusion is essential to summary the main finding and results.
Comments on the Quality of English Language
Minor editing of English language required.
Reviewer 3 Report
Comments and Suggestions for Authors
In this review authors address in a light and superficial way the use of brain-computer interface on Optogenetics. The paper is easy to read and all main issues are addressed in introductory and basic way. Neverthless the paper may be useful to newcommers to the field. I would just ask authors to go a bit deeper on Chapter 2.3 "Stimulus system" and in general on the optics problems challenges and potential. A few typos can be easily corrected. Please remove line 555.
Comments on the Quality of English Language
Quality is fine. Just a few typos easily corrected
Reviewer 4 Report
Comments and Suggestions for Authors The title to the article is very attention catching. If prepared and written in the correct way, it can easily become a highly cited paper. However, the authors present an article that resembles a Chat GPT written paper. The format and content in the paper in the current format is dry and boring, not much insightful. For a review, the manuscript needed to be more in depth, long, clearly, with more schematics, and more insightful and detailed number of references. If accepted for publication, the authors will first need to improve the paper in the highlighted aspects. Comments on the Quality of English Language
The current article shape feel like a Wikipedia, Chat-GPT/Gemini/Facebook Llama written paper.
Round 2
Reviewer 1 Report
Comments and Suggestions for Authors
The main goal of the manuscript is to review the development and application of optogenetic BCIs. It provides an overview of existing literature on optogenetic BCIs.
General comments
The revision improves the overall quality of the text, making it clearer and more readable.
However, the manuscript can still be improved considerably. First, the manuscript lacks a clear structure, or systemization of the topic(s). Second, it remains unclear what the contribution and novelty is, as the manuscript does not identify a knowledge gap.
Specific comments
39 “to improve the brain's ability to control BCIs [5-8]” – Ref. 5 is superfluous, as ref. 6 makes an explicit claim about BCI performance in the exact same setting.
45 “These techniques can be broadly categorized into two main types” – It seems that with ref. 7, you introduced a third type, namely ultrasound.
371 “Table 1. Applicability and advantages of each system.” – The table caption should be more descriptive, such that it can be understood on its own. Additionally, I am unsure how to interpret the “Part” and “Advantages” columns of the table. For example, it is not clear why “Video signals combined with electrical signals” is an advantage. Lastly, the correspondence between the works presented in table and those discussed in the text is unclear, as “Bertram et al (1997)” is seen as “[47]” in the text. Therefore, consider including the reference number, e.g. [47], in the “Reference” column of the table.
Comments on the Quality of English Language
The manuscript could benefit from some editing by a qualified person.
Reviewer 3 Report
Comments and Suggestions for Authors
Authors provided a good revision of the paper taking into account the comments made. The paper can be accepted in current form.
Author Response
Thank you very much for your time and effort on our behalf. We also greatly appreciate your recognition and support.
Reviewer 4 Report
Comments and Suggestions for Authors
This is an in-depth, high-quality article that has been significantly enhanced by the authors' thorough response to the initial round of reviews. The revised paper boasts robust technical details, offering insightful and practical content that makes it a valuable resource for both theoretical and technical reference. The authors have successfully struck a balance between providing essential theoretical context and offering practical details that will facilitate implementation and investigation. I recommend this paper for publication in its current form, as it is now a comprehensive and insightful contribution to the field.
Author Response
Comments 1: This is an in-depth, high-quality article that has been significantly enhanced by the authors' thorough response to the initial round of reviews. The revised paper boasts robust technical details, offering insightful and practical content that makes it a valuable resource for both theoretical and technical reference. The authors have successfully struck a balance between providing essential theoretical context and offering practical details that will facilitate implementation and investigation. I recommend this paper for publication in its current form, as it is now a comprehensive and insightful contribution to the field.
Response 1: Thank you very much for your time and effort on our behalf. We also greatly appreciate your recognition and support.
Round 3
Reviewer 1 Report
Comments and Suggestions for Authors
No further comments.
Comments on the Quality of English Language
line 55 (and more) - "Ultrasound(tUS)" vs "deep brain stimulation (DBS)" ~ Inconsistencies in spacing
line 67 - "..., Optogenetic stimulation" ~ Capitalization error
I won't make a list of all typo's, just re-read the text attentively to catch these types of inconsistencies and errors...